# The Potential of Mesenchymal Stem/Stromal Cells in Diabetic Wounds and Future Directions for Research and Therapy—Is It Time for Use in Everyday Practice?

**DOI:** 10.3390/ijms252212171

**Published:** 2024-11-13

**Authors:** Damian Sieńko, Ilona Szabłowska-Gadomska, Anna Nowak-Szwed, Stefan Rudziński, Maksymilian Gofron, Przemysław Zygmunciak, Małgorzata Lewandowska-Szumieł, Wojciech Stanisław Zgliczyński, Leszek Czupryniak, Beata Mrozikiewicz-Rakowska

**Affiliations:** 1Department of Diabetology and Internal Diseases, Medical University of Warsaw, 02-097 Warsaw, Poland; damiansienko7@gmail.com (D.S.); anna.nowak@wum.edu.pl (A.N.-S.); bigosik@poczta.onet.pl (L.C.); 2Doctoral School, Medical University of Warsaw, 02-091 Warsaw, Poland; 3Laboratory for Cell Research and Application, Medical University of Warsaw, 02-097 Warsaw, Poland; ilona.szablowska-gadomska@wum.edu.pl (I.S.-G.); stefan.rudzinski@wum.edu.pl (S.R.); malgorzata.lewandowska-szumiel@wum.edu.pl (M.L.-S.); 4Department of Urology, Municipal Complex Hospital, 42-200 Czestochowa, Poland; max9615@gmail.com; 5Department of Endocrinology, Centre of Postgraduate Medical Education, Bielanski Hospital, 01-809 Warsaw, Poland; zygmunciakprzemyslaw@gmail.com (P.Z.); zgliczynski.w@gmail.com (W.S.Z.); 6Department of Histology and Embryology, Medical University of Warsaw, 02-004 Warsaw, Poland

**Keywords:** mesenchymal stem/stromal cells, cell therapy, diabetic foot ulcer, allogenic cell therapy, ATMP

## Abstract

The treatment of diabetic wounds is impaired by the intricate nature of diabetes and its associated complications, necessitating novel strategies. The utilization of mesenchymal stem/stromal cells (MSCs) as a therapeutic modality for chronic and recalcitrant wounds in diabetic patients is an active area of investigation aimed at enhancing its therapeutic potential covering tissue regeneration. The threat posed to the patient and their environment by the presence of a diabetic foot ulcer (DFU) is so significant that any additional therapeutic approach that opens new pathways to halt the progression of local changes, which subsequently lead to a generalized inflammatory process, offers a chance to reduce the risk of amputation or even death. This article explores the potential of MSCs in diabetic foot ulcer treatment, examining their mechanisms of action, clinical application challenges, and future directions for research and therapy.

## 1. Introduction

The global incidence of diabetes mellitus (DM) in individuals aged 20–79 years was estimated at 10.5%, encompassing approximately 536.6 million individuals in 2021. This figure is projected to rise to 12.2%, affecting an estimated 783.2 million by 2045 [1]. A prevalent complication of DM, diabetic foot ulcer (DFU), afflicts up to one-third of patients over their lifetime, leading to substantial morbidity characterized by challenging treatment courses and a high recurrence rate.

The management of chronic and recalcitrant wounds in diabetic patients remains a significant challenge in contemporary medicine. The intricacies of wound healing in diabetic patients, compounded by impaired angiogenesis, delayed re-epithelialization, and diminished collagen deposition, contribute to the chronic nature of these wounds [2,3]. In light of the constraints inherent in conventional therapeutic modalities for diabetic wound care, contemporary research has increasingly pivoted toward exploring the potential of mesenchymal stem/stromal cells (MSCs) as a novel therapeutic approach in this domain. MSCs are extensively recognized for their crucial role in immunomodulation, hematopoiesis, and tissue repair. Due to these distinctive characteristics, they have gained rapid recognition as a promising therapeutic agent in the field of regenerative medicine [4,5]. MSCs are multipotent cells capable of differentiating into adipocytes, osteoblasts, and chondrocytes. They are identified by positive expressions of CD73, CD90, and CD105, and the absence of hematopoietic markers like CD34 and CD45. MSCs can be derived from various tissues, including bone marrow and adipose tissue, and exhibit heterogeneity based on their origin. In clinical contexts, pure MSCs refer to isolated, well-characterized cell populations that meet specific phenotypic and functional criteria. In contrast, MSCs in the stromal vascular fraction (SVF) are part of a mixed cell population, containing other cell types like endothelial and immune cells. This distinction is critical for interpreting clinical outcomes, as pure MSCs offer targeted effects, whereas SVF-derived therapies involve multiple cell interactions, potentially complicating the understanding of therapeutic efficacy [6,7]. MSCs, particularly those derived from sources like umbilical cords and adipose tissue, have shown promise in enhancing wound healing through differentiation and angiogenesis [3,8]. In the context of diabetic foot ulcers (DFUs), MSCs exhibit the capacity to modulate various critical processes essential for efficacious wound healing. As demonstrated by randomized clinical trials, stem cells are capable of improving the treatment of DFU, reducing its cost, and additionally decreasing the frequency of hospital admissions for patients with this complication [9,10,11,12]. These processes include the modulation of inflammatory responses, the synthesis of the extracellular matrix, the facilitation of keratinocyte migration, and the promotion of angiogenesis [13,14,15]. In pursuit of augmenting the therapeutic efficacy of MSCs, several innovative methodologies are being employed. These include preconditioning of MSCs, genetic modification, the application of combination therapies, and the utilization of exosome-based approaches [16]. For instance, recent advancements in elucidating the role of MSC-derived extracellular vesicles (EVs) have unveiled novel therapeutic pathways. These developments promise substantial benefits in the realms of tissue repair and regeneration, particularly in the treatment of diabetic wounds [17].

Despite significant progress in delineating the pathophysiology of DFUs and associated cellular and molecular mechanisms, there remains a notable deficit in efficacious treatment modalities. Present therapeutic approaches are often ineffectual in expeditiously healing deep, chronic wounds, particularly those complicated by microvascular obstruction. This challenge is further exacerbated by the frequent recurrence of DFUs, coupled with their correlation to elevated mortality rates and substantial healthcare expenditures. The current evidence underpinning much of the routine clinical management of DFUs is, regrettably, limited. This paucity of robust evidence underscores the imperative need for enhanced clinical trials and systematic evaluations of routine care outcomes, particularly across diverse health economies [18,19].

This manuscript endeavors to investigate the potential applications of MSCs in the therapeutic management of DFUs. It aims to elucidate the underlying mechanisms of action of MSCs, delineate the challenges associated with their clinical application, and forecast future trajectories for both research and therapeutic interventions in this domain.

## 2. Pathophysiology of Impaired Wound Healing in Diabetes Mellitus

Impaired wound healing in diabetes results from several overlapping processes, including endotheliopathy, neuropathy, and immunopathy.

### 2.1. Endoteliopathy

Endotheliopathy plays a critical role in the dysfunction of wound healing observed in patients with diabetes mellitus, acting as a fundamental mechanism that underpins the complex interplay between vascular, neuropathic, and metabolic disturbances characteristic of this condition. The pathological changes initiated by endothelial dysfunction in diabetes include both microvascular and macrovascular alterations, which collectively impair blood flow, reduce nutrient and oxygen delivery to the wound site, and hinder the removal of metabolic waste products [20]. Research underscores that the persistent hyperglycemic state characteristic of type 2 diabetes mellitus initiates non-enzymatic glycation reactions with proteins, lipids, and nucleic acids, leading to the formation of advanced glycation end products (AGEs). These AGEs are central to the progression of diabetic complications, contributing to oxidative stress and inflammation through various mechanisms, including the activation of the receptor for AGEs pathway. This process, in turn, exacerbates endothelial damage and impairs the natural angiogenic response necessary for tissue repair and vascular health [21,22,23]. Furthermore, hyperglycemia-induced metabolic derangements in endothelial cells, as evidenced by experimental models, lead to increased endothelial cell layer permeability and decreased capillary density. These changes underline the vascular dysfunction integral to diabetic cardiomyopathy, emphasizing the role of chronic hyperglycemia in inducing oxidative stress within diabetic endothelial cells [24]. This understanding is pivotal for developing targeted therapeutic interventions aimed at mitigating these vascular complications. 

Research indicates that in diabetic patients, impaired immune response significantly hampers wound healing. Diabetic foot ulcers show a lack of necessary immune cell recruitment for normal healing processes. This deficiency points to potential targets for future therapeutic interventions, highlighting the importance of understanding how the healing process is disrupted in diabetic wounds to design more effective treatments [25]. Collectively, these factors underscore the critical role of endotheliopathy in the multifaceted process of wound healing dysfunction in diabetes mellitus, highlighting the need for targeted therapeutic strategies to mitigate vascular complications and enhance wound repair. Recent studies suggest MSC-based therapy may have a positive effect on the process of endotheliopathy by reducing endothelial cell damage and inflammation, thus improving vascular function in diabetic patients [26].

### 2.2. Neuropathy

Diabetic neuropathy (DN) is a prevalent and debilitating complication of diabetes mellitus, affecting up to 51% of patients [27]. Recent studies from 2021 onwards continue to support the understanding that chronic hyperglycaemia, activation of the polyol pathway, increased oxidative stress, and the formation of advanced glycation end products (AGEs) are central to the pathogenesis of diabetic neuropathy. Research provides further insights into these contributing factors. The activation of the polyol pathway, a consequence of persistent hyperglycaemia, contributes significantly to nerve damage through the accumulation of sorbitol and fructose, leading to osmotic and oxidative stress within nerve cells [28,29]. Oxidative stress, heightened by chronic hyperglycaemia and acute glucose fluctuations, plays a significant role in endothelial dysfunction and diabetic neuropathy, indicating its role in the deterioration of nerve function [30,31]. Additionally, the formation of AGEs through non-enzymatic reactions between proteins and reducing sugars contributes to the structural and functional alterations of tissues, exacerbating neuropathic conditions in diabetes [32,33]. These above-mentioned mechanisms collectively underscore the complex nature of diabetic neuropathy’s development, highlighting the importance of controlling hyperglycaemia and addressing oxidative stress as part of comprehensive management strategies to mitigate the progression of this debilitating complication (Figure 1). In addition to hyperglycaemia, factors including fluctuations in blood sugar levels, disturbances in lipid profiles, smoking, and excessive alcohol consumption can also affect the presence and intensity of neuropathy symptoms [34]. Damage to the vasa nervorum, the blood vessels that nourish diverse nerve fibers including C-type and delta fibers vital for nervous system functions, is primarily linked to the hindered healing of wounds in diabetes mellitus, particularly recurring issues in diabetic foot ulcers [35]. Recent research has shown that MSC-based therapy can significantly influence neuropathy by promoting nerve regeneration and reducing inflammation. MSCs release exosomes that carry proteins, microRNAs, and other factors, which can suppress inflammatory pathways and enhance nerve repair in diabetic neuropathy. These therapeutic effects have been observed in both animal models and preclinical studies, where MSCs improved nerve function and reduced neuropathic pain, suggesting a promising avenue for treating peripheral neuropathy in diabetic patients [36].

### 2.3. Immunopathy

Diabetic immunopathy refers to the complex alterations in the immune system resulting from diabetes mellitus, contributing to a wide array of complications that significantly impact patient morbidity and mortality. The pathophysiology of diabetic immunopathy is multifaceted, involving again hyperglycemia-induced oxidative stress, chronic inflammation, and immune dysregulation, which collectively predispose individuals to infections, impaired wound healing, and accelerated atherosclerosis. Hyperglycemia plays a central role in diabetic immunopathy by promoting oxidative stress and the production of advanced glycation end products (AGEs) [33]. These processes lead to endothelial dysfunction and a pro-inflammatory state, characterized by increased circulating levels of inflammatory cytokines such as tumor necrosis factor-alpha (TNF-α) and interleukin-6 (IL-6) [37]. Chronic inflammation significantly contributes to the dysregulation of both innate and adaptive immunity, impairing neutrophil function and reducing the proliferative response of lymphocytes, thereby compromising the body’s ability to combat infections and heal wounds effectively [38,39,40]. In another study, proteomic analysis demonstrated the presence of proteins that may be responsible for mitigating inflammatory processes in patients with diabetic foot syndrome undergoing allogeneic MSC therapy [26]. The alteration of the immune system in diabetic individuals also exacerbates the severity of viral infections, such as SARS-CoV-2, dengue, and influenza, compared to healthy individuals. The dysregulated immune response in diabetes leads to heightened susceptibility and escalated disease severity, emphasizing the critical need for effective management of diabetes to mitigate these adverse outcomes. Moreover, diabetic immunopathy contributes to the pathogenesis of microvascular and macrovascular complications. The immune-mediated inflammatory processes accelerate the development of diabetic retinopathy, nephropathy, and neuropathy, with diabetic nephropathy being particularly influenced by the role of innate immunity in promoting inflammation and tissue fibrosis, ultimately leading to renal failure [41,42,43]. This illustrates the significant impact of immune dysregulation on the progression of diabetes-related complications.

The underlying pathophysiology of diabetes, including impaired angiogenesis, chronic inflammation, and reduced neovascularization, often compromises wound healing in diabetic foot syndrome [44].

## 3. MSC Role in Wound Healing Phases

MSCs have emerged as a promising therapeutic strategy to enhance wound healing in DFU, leveraging their multifaceted role in the wound healing processes, including inflammation, proliferation, and tissue remodeling [45,46].

### 3.1. Inflammatory Phase

In diabetic patients, the inflammatory phase of the wound healing process is often prolonged, leading to chronic wound occurrence. MSCs have the potential to react to the inflammatory environment affecting various molecular pathways. In vitro studies have demonstrated that MSCs can reduce the expression of pro-inflammatory cytokines such as tumor necrosis factor (TNF)-α and interleukin (IL)-1β [47]. They have also shown that MSCs secrete anti-inflammatory cytokines, including interleukin-10 (IL-10) and transforming growth factor-beta (TGF-β), which modulate T-cell responses by suppressing excessive Th1 responses and promoting a shift toward a Th2-type response [48,49]. Another pivotal role of MSCs is the reduction in reactive oxygen species (ROS), which are elevated in diabetic wounds. By regulating ROS levels, MSCs mitigated oxidative stress-induced damage in the murine model, facilitating the resolution of inflammation [50]. Additionally, Cho et al. demonstrated, using an in vitro co-culture system with mouse bone marrow-derived macrophages, that MSCs modulated macrophage polarization, shifting them from a pro-inflammatory (M1) to an anti-inflammatory (M2) phenotype, which supported healing and reduced chronic inflammation [51,52]. All those actions lead to the faster transition from the inflammatory phase to the proliferation phase of wound healing.

### 3.2. Proliferation Phase

During the proliferation phase, MSCs contribute to tissue formation and repair by promoting the proliferation of fibroblasts, keratinocytes, and endothelial cells, as well as enhancing angiogenesis. Throughout this process, MSCs release paracrine growth factors that facilitate tissue regeneration and promote wound healing [45]. Saheli et al. investigated using MSC-conditioned media (MSC-CM) in enhancing wound healing in diabetic conditions. By using both in vivo diabetic rat models and in vitro high-glucose fibroblast cultures, researchers observed improved wound closure rates, reduced inflammation, better tissue remodeling, and increased vascularization in MSC-CM-treated wounds. Gene analysis showed upregulation of growth factors epidermal growth factor (EGF) and basic fibroblast growth factor (bFGF), indicating that MSC-CM promotes fibroblast viability, proliferation, and migration [53]. Yates et al. showed that MSC treatment supports fibroblast function by promoting their survival and movement, which leads to enhanced extracellular matrix (ECM) deposition, boosting overall healing effects in the in vivo murine model [54]. Additionally, Li et al. demonstrated that exosomes derived from MSCs play a crucial role in stimulating fibroblast proliferation and migration, further aiding the wound repair process [55]. At the same time MSCs enhance neovascularization by secretion of growth factors such as vascular endothelial growth factor (VEGF) crucial for new blood vessel formation and wound closure. Wu et al. conducted a study using genetically diabetic db/db mice and discovered that lesions treated with MSCs exhibited significantly increased levels of VEGF, Angiopoietin-1, and keratinocyte-specific proteins. The MSCs enhanced keratinocyte proliferation, stimulated angiogenesis, promoted epithelial regeneration, and accelerated overall wound healing [2,56]. Moreover, in the context of chronic wounds associated with diabetes, the application of a synthetic three-dimensional collagen scaffold infused with VEGF has been shown to improve blood vessel formation and accelerate wound healing in diabetic murine model [57].

### 3.3. Remodeling Phase

The remodeling phase focuses on the scar’s maturation and achieving optimal tensile strength through the rearrangement, breakdown, and renewal of the ECM. In this phase, the density of collagen fibers within the granulation tissue increases as part of the tissue structure’s restoration [58]. During the remodeling phase, MSCs modulate the activity of matrix metalloproteinases (MMPs) and their inhibitors (TIMPs), balancing the breakdown and synthesis of ECM components and allowing tissue remodeling [59]. Lozito et al. demonstrated this regulatory effect in an in vitro model, showcasing MSCs’ capacity to modulate ECM dynamics, essential for effective wound healing and tissue restructuring [60]. Similarly, Li et al. revealed that MSC-CM can significantly modulate ECM remodeling by upregulating mRNA levels of matrix metalloproteinases MMP-2 and MMP-9, while downregulating tissue inhibitors TIMP-1 and TIMP-2, thereby promoting the wound healing remodeling phase [61]. Additionally, MSC-CM activates the Erk signaling pathway, especially in diabetes-mimicking environments, to enhance keratinocyte proliferation and migration [61]. These in vitro findings emphasize MSCs’ therapeutic potential in regulating ECM dynamics and support MSC-based therapies to improve tissue repair and regeneration.

Reassuming MSCs offer a promising therapeutic approach for the regeneration of injured tissues in patients with diabetes mellitus, especially with diabetic foot ulcer. Their ability to modulate the inflammatory response, promote tissue proliferation and enhance vasculature regeneration addresses the multifaceted challenges in healing diabetic foot ulcers (Figure 2). However, clinical application requires further research to optimize MSC delivery methods and dosages and understand their mechanisms of action in diabetic foot syndrome.

## 4. The Legal Regulations Regarding Cell-Based Products

Researchers who explore the properties of mesenchymal stem/stromal cells (MSCs) to apply them in clinical practice must navigate a challenging process to prepare these cells for administration to patients. Below, we describe the regulatory issues governing the clinical application of cell-based products. These include compliance with stringent regulations on cell isolation, cultivation, quality control, and safety measures, as well as obtaining the necessary approvals from relevant health authorities to ensure the product’s safety and efficacy for therapeutic use. The cell-based medicinal products and related procedures are regulated by law and are considered Advanced Therapy Medicinal Products (ATMP) [62]. Throughout the European Union, there are three main categories of ATMPs. First, somatic-cell therapy medicines (also called somatic Cell Therapy Medical Products- sCTMP) are products containing manipulated cells to change their biological properties or be used for purposes other than their primary function in the human body. The second group of ATMP is called tissue-engineered medicines (TEM) or products (TEP), and it contains cells that are modified in such a way that they can be used to repair, regenerate, or replace diseased or damaged tissues. The third category includes gene therapy medicines (GTM, also called gene therapy medicinal products—GTMP), which are increasingly popular nowadays. These involve introducing laboratory-recombined genes into a patient’s body, such as Gene Therapy Medicinal Products (GTMP), for example, Chimeric Antigen Receptor T-cells (CAR-T). There are also combined ATMPs with one or more medical devices, e.g., cells seeded on a scaffold [62,63,64]. In the EU, the first ATMP product appeared on the market in 2009, and, to date, at least 24 products have been approved by a regulatory body [65]. In the US there are only two categories of this type of product: Cellular and Gene Therapy Products (accessed on 16 September 2024) [66]. Currently, there are 38 licensed cellular and gene therapy products (accessed on 16 September 2024) [66]. Overall, ATMP constitute only a small part of the pharmaceutical market and are mostly developed in academic centers [67].

Not every new ATMP medicinal product will have a chance to appear on the market. Some products will not successfully pass the preclinical phase, others will stop at the early phases of clinical trials. Only a few, such as Alofisel, containing expanded ADSC, will complete the registration process. It illustrates the wider problem concerning the ATMPs’ development. First of all, the requirements of the regulatory bodies are identical, i.e., equally demanding for the different stages of the product development. Therefore, validation of each production stage must be carried out in accordance with many strict regulations at a very early stage of ATMP development. In the case of the standard medicine, there are fewer regulations to be watched and the general methodology is, in principle, the same—typical for chemical substances. In the case of advanced therapy medicinal products (ATMPs), the validation plan is in most cases individual, which is associated with a significantly greater investment of time and costs and is always associated with a longer path to regulatory approval. It is also burdened with the risk of additional requirements. Furthermore, the EMA recommendations dedicated to ATMPs change dynamically. It should be mentioned that the “Guideline on quality, non-clinical and clinical requirements for investigational advanced therapy medicinal products in clinical trials” (EMA/CAT/123573/2024), currently required by the Clinical Trials Registration Offices, was announced on March 2024 as a draft with no date for coming into effect, while the previous version, from January 2019 (EMA/CAT/852602/2018), remains at the draft stage, also with no date for coming into effect. Those guidelines are accompanied by many other obligatory recommendations—frequently changing and not always entirely consistent with each other. Taking into account, that because of the species-determined active substance, the preclinical proof of concept based on animal observations is far from the real situation, as pharmaceutical companies are not interested in investing in ATIMP development during the early stages of clinical trials. Therefore, in this phase, the financial and organizational burden falls on the shoulders of the research centers, which is a serious obstacle to ATIMP being introduced to the market.

After entering the keyword “cell therapy” the clinicaltrials.gov website searches for 8266 completed in phases spanning from early phase 1 to phase 4 (Figure 3). After introducing an additional filter, “diabetic foot” only 24 trials were found (as of 24 October 2024) (Table 1).

There are two legislative pathways regulating the application of ATMP to patients. The first describes the use of ATMP in a clinical study (also called Advanced Therapy Investigational Medicinal Product when undergoing clinical study—ATIMP). The second is the use of ATMP in a hospital-exemption (ATMP-HE) route where the ATMP is prepared on request, under the responsibility of the doctor, according to the individual needs of the patient (Figure 4).

The estimated time that may pass from the idea to launching the new product on the market lasts on average 15 years and it consumes large amount of financial resources [67,68].

It was expected that, due to their properties, cell-based products will enable the treatment of patients for whom current therapies cannot bring results or their effects are unsatisfactory. Regulatory authorities are aware of the unmet needs of the health market and are taking steps to accelerate the evaluation of new drugs. However, the legal requirements for ATMP consist of a plethora of documents and regulations, many of which are often intertwined. This leads to ambiguous interpretations hindering the development and approval processes.

Production of ATMP takes place in the pharmaceutical factory (PhF) that meets the requirements of the directive that regulates the good manufacturing practice (GMP) rules in respect of medicinal products for human use [69]. However, a pharmaceutical factory cannot accept biological material as a starting material for ATMP production directly from a recruiting center (clinic). Such a material has to go through the Tissue and Cell Bank (TCB) where it is qualified and accepted for further use (Figure 4). In Poland, entities that own ATMP factories are sometimes also registered as Tissue and Cell Banks and are therefore subjected to the regulations of European Commission: 2004/23/EC (directive on tissue and cells banking) [70], 2002/98/EC (directive on human blood banking) [71], or both, as well as the national regulatory system Biological material, e.g., adipose tissue, is verified twice, first in TCB and then in PhF. Then, after the positive qualification in PhF, the biological material (e.g., adipose tissue) is subjected to the cell isolation process (usually using the enzymatic method). After isolation, the final cell-containing product can be prepared in a straight line, or cells are subjected to other manipulations. Under the regulations, even if nothing more than cell culture is carried out for the purpose of increasing a cell population to obtain a desired cell resource, it is considered “substantial manipulation”. Potential more invasive interference, like gene transduction, may require a different, separate infrastructure. In the case of MSCs intended to be used in wound healing, there are no genetic manipulations proposed so far. However, the stage of cell culture is obligatory for MSCs isolated from adipose tissue not only because of the desired cell expansion but also in order to obtain the pure MSC immunophenotype [72]. During the ATMP manufacturing process, there are many steps where samples are acquired to perform quality control tests (e.g., sterility, determining the level of endotoxins, excluding mycoplasma). For a more detailed description of the manufacturing process using ADSC and quality control steps see Szabłowska-Gadomska et al., 2023 [73].

To sum up, there are several regulatory bodies involved in the process of making MSC-based products available for the treatment of diabetic wounds. On the one hand, it makes the road to the patient complicated, costly, and time consuming. On the other hand, it is intended to ensure the safety of the treatment. Importantly, we are observing the growing understanding of the specificity of cell-based products by regulatory bodies. Especially, agencies such as FDA or EMA are constantly working on making regulations more and more adequate. It seems that the evidently growing interest in cell-based therapies for various clinical situations will guarantee the expected progress—for the benefit of their use in diabetic foot ulcer as well.

## 5. Allogeneic Products’ Advantages and Challenges

The rising research focus toward allogeneic applications in therapies using mesenchymal stem/stromal cells isolated from adipose tissue seemed to be a natural consequence of the already known positive effects of using allogeneic mesenchymal stem/stromal cells isolated from bone marrow. In immunological terms, mesenchymal stem/stromal cells belong to a special group of cells because they do not express or have significantly reduced expression of MHC class I and II and co-stimulatory molecules (CD40, CD80/CD86) [74]. According to the clinicaltrials.gov website (keyword allogeneic MSC), 169 clinical trials using MSC in an allogeneic system are being conducted worldwide (accessed on 16 September 2024). There are also products available on the medical market containing allogeneic mesenchymal cells isolated from adipose tissue, e.g., Alofisel. Currently, more studies where MSC-based drug products are subsequently administered in an allogeneic setting are conducted.

The path to development of the allogenic MSC therapies is related to their described and extensively studied immunomodulatory properties [75]. ATMPs based on allogeneic material provide a chance to manufacture the product even when the recipient-patient’s health prevents the collection process, as may happen in the case of autologous products. Even a relatively minimally invasive procedure such as liposuction may pose a risk for patients with tendencies for the occurrence of chronic wounds (e.g., patients suffering from diabetes).

Allogeneic ATMPs also have a greater potential to be available in a shorter time, almost immediately. There is no need to wait for a long process of cell isolation and multiplication because they can be performed even before qualifying the recipient-patient for treatment.

The allogeneic system makes it possible to obtain cells from waste material, e.g., after liposuction, which would be of no use for donor-patients. Such material from healthy donors, after positive qualification, can be a valuable source of cells or a secretome for patients with health deficits.

However, despite the ever-growing research interest in MSC-based therapies, from a PhF’s point of view, the allogenic ATMPs pose quite a challenge. In case of direct autologous transplantations (without isolation, manipulation, etc.), in Poland, the acceptance criteria are well-defined and classified by national regulatory system, whereas for ATMP production (especially allogenic) the biological material has to undergo additional selection according to criteria based on Directive 2004/23/EC—Setting standards of quality and safety for the donation, procurement, testing, processing, preservation, storage and distribution of human tissues and cells [70].

Therefore, in addition to serological tests, physical examination and diagnostic tests are performed to exclude the most common cancer diseases, e.g., breast cancer, prostate cancer, uterus cancer, and lung cancer. The latter are also indicated by cancer marker tests. This approach toward potential donors of biological material may be seen as an advantage as it provides access to a wider panel of tests that are not routinely performed diagnostic tests but that have a significant preventive effect.

Regardless of whether therapies based on the administration of live cells or their secretome will ultimately prove to be the most effective therapies, the possibility of using an allogeneic system is critically important, especially in applications in diabetic patients.

## 6. Practical Issues Related to ATMP Release, and Its Delivery from the PhF to the Patient

A short shelf-life and strict requirements for storage and transport conditions of ATMP intended for immediate application after production pose a serious challenge for manufacturers and clinicians. The expiration date of such products varies and depends on the specificity of the cells themselves as well as the excipient in which the cells are suspended. Most often, the shelf-life spans a dozen to several dozen hours from the time the products are manufactured. This is especially important when the distance between the PhF and the clinic is long. A solution may be to transport deep-frozen products and defrost them in the clinic at the patient’s bedside. However, this method requires the use of cryoprotectants (e.g., DMSO), which may sometimes pose an additional risk to patients. In addition, the condition of freshly thawed cells may be worse, compared to those obtained directly from the culture dish (for purely biological reasons) and may result in a weakened therapeutic effect. François et al. showed that the percentage of viability MSC collected from continuous culture was about 90%, while their freshly thawed counterparts dropped to about 40%. They also found a deterioration in the immunosuppressive properties of the tested cells [76]. Some authors even suggest that the poorer properties of the cells after thawing may be the reason for the unsatisfactory results of clinical trials with their participation [77,78,79]. Therefore, it is recommended to consider this important aspect during the planning of the manufacturing process, creation of the new product specification, and later during the testing of cell parameters at the stage of application of the finished product. This will allow us to avoid the high costs of clinical trials due to suboptimal use of the potential of MSC cells during manufacturing and the risk of interference resulting from the impact of the thawing process itself.

The use of a cell secretome may be a cheaper and logistically easier solution due to feasible normalization of the storage conditions [80]. MSCs can secrete substances capable of, among others: regulating the functions of immune system cells, and improving the healing microenvironment [81]. Medicinal products containing cell secretome may be perceived as safer from a microbiological point of view versus cells because they could be sterilized (e.g., by filtration). Additionally, it would be easier to standardize the secretome in terms of the tested parameters, e.g., dose and potency. However, the use of cells has the advantage that their secretome may vary depending on changing environmental conditions [82]. Therefore, a specific cell response to the local environment of the wound might be expected and result in secreting a specific and the most required cocktail of factors.

In the case of medicinal products with live cells, it is permissible and is often the practice to administer them to the patient before receiving all the results of analytical tests, including the initial certification and release for administration. Then, the final certification is carried out by a Qualified Person (QP) after acquiring all test results. This is due to the required sterility tests, which according to pharmacopoeial methods require 14 days of incubation, while the shelf life of the product may be several hours. More information on this subject can be found in the work of Szabłowska-Gadomska et al. [73].

The biological properties of MSCs are difficult to assess in animal models, especially when considering the immunomodulatory properties as their main therapeutic value [81,83]. Using the immunocompromised animals required for human MSC testing, to assess the dosage of MSCs while preparing the ATMP, e.g., for DFU treatment, may result in a loss of information regarding the immunomodulatory properties of MSCs and its relation to the dosage. Thus, ATMPs containing live MSCs face a critical issue of dosage selection. In clinicaltrials.gov, the variety of doses used in registered clinical studies ranges from 2 to 100 × 10^6^/mL applied both systemically or locally per injection or directly on wounds (accessed on 19 February 2024).

One of the concerns expressed by regulatory bodies is the supposed cancerogenic potential of MSC. Therefore, EMA guidelines were issued in 2024 regarding the requirements for preclinical ATIMP studies involving an extensive tumorigenic and immunogenic safety assessment (EMA/CAT/123753/2024). However, MSCs show almost negligible immunogenicity and by some are even called immune-evasive [75,84]. Moreover, unmodified MSCs, that are often used in cell therapies, which underwent only minimal in vitro manipulations such as isolation and expansion retain their genetic stability. The reported rare tumorigenic events were not caused by MSC but by the ATIMP contamination with tumor cells [85]. Therefore, legal bodies have applied an especially high pressure on researchers to ascertain the identity of cells used in ATIMP as the origin and purity of starting cellular material is one of the key factors in the safety of the ATIMP.

## 7. The Ways of Administration of MSC to Intended Locations

The utilization of MSC as a therapeutic modality for chronic and recalcitrant wounds in diabetic patients is an active area of investigation aimed at enhancing its therapeutic potential and expediting tissue regeneration. The method of delivering MSC to the intended location of action is a key topic of scientific interest. Through a comprehensive review of existing literature and clinical studies we can distinguish two main ways of administering MSC, which are a systemic delivery and a local administration. 

Local administration includes the direct injection of MSC into the wound alone or the inclusion in a collagen sponge, on hydrogel scaffolds, or on other specially designed platforms.

Systemic delivery is divided into intravenous and arterial administration [45].

By investigating the complex differences between the mentioned two methods, we intend to analyze each modality’s safety record, therapeutic utility, and effectiveness in relation to find the proper administration method for chronic wound therapy.

The utilization of specialized scaffolds as a platform for the local delivery of the stem cells has emerged as a promising therapeutic approach that has garnered significant attention in the scientific community and clinical trials. Regardless of the constituent materials utilized, the shared objective of these scaffolds is to promote tissue repair, improve stem cell survival, augment cell distribution within the wound and support their secretory functions [86]. The most advantageous modalities appropriate for the chosen biomaterials have been carefully determined and selected by the authors as part of the scope of this review.

Hydrogel scaffolds have emerged as a promising approach in regenerative medicine due to their biodegradability, ability to promote cellular survival, and controlled release of drugs and biomolecules [86,87]. Furthermore, hydrogel can be developed into a three-dimensional printing nano-architecture to create a sustainable and attractive stem cell niche, ensuring proper micro-structure and support. The three-dimensional extracellular matrix created by the Pluronic F-127 hydrogel exhibits excellent cellular affinity, making it a supportive environment for stem cell proliferation and it has been frequently utilized as a graft medium in cellular therapies with a high rate of success [88]. Collagen fibrils in hydrogels, which contain Arg-Gly-Asp (RGD) sequences, provide binding sites for cell interaction and adhesion [89]. Following this, MSC bound to the scaffolds can be minimally invasively injected at the site of a lesion, reducing the need for surgical intervention. Moreover, hydrogel scaffolds are capable of loading hydrophilic drugs and biomolecules, and fibrin hydrogel has been proposed as a means of improving the retention and stability of BMSCs-exos in vivo [90]. However, one of the drawbacks associated with hydrogel scaffolds is that the release of drugs can be problematic, as uncontrolled diffusion, or unfavorable conditions, can lead to inadequate kinetics and delivery. In particular, molecules with low steric hindrance may diffuse uncontrollably, making them difficult to manage with precision. Additionally, the loading of hydrophobic drugs with limited affinity for the aqueous environment presents a significant limitation. However, it is possible to address these issues by incorporating bonds between drug molecules to regulate the rate of release. Depending on the strength of these bonds, they can be broken more or less easily, resulting in a more controlled delivery of stem cell biofactors.

In a study conducted by Shi et al., a chitosan/silk hydrogel was used in an in vivo animal setting. The resulting scaffold demonstrated desirable properties, including appropriate moisture retention and swelling characteristics. Studies have demonstrated the potential of GMSC-derived exosomes transferred in the environment of a chitosan/silk hydrogel in promoting wound healing in diabetic patients. The addition of GMSC-derived exosomes resulted in a significant reduction in wound size compared to both the control group and the group treated with hydrogel alone, indicating the potential of exosomes as a therapeutic agent for diabetic wound healing. Furthermore, it was observed that the group treated with hydrogel alone exhibited a significantly smaller wound size at both 1 and 2 weeks of treatment compared to the control group, highlighting the potential of hydrogel as a treatment option for wound healing [91].

Overall, hydrogel scaffolds offer a promising avenue for regenerative medicine, with further research needed to address the limitations and optimize their potential.

A widely used strategy in scaffold design is to utilize materials that can stimulate MSC and boost their secretory functions by modulating their molecular behavior. This category of materials is quite extensive, and for the purpose of this study, we selected the most promising candidates. Among these candidates is the gelatin-sericin (GS) scaffold, which is coated with laminin (GSL) and a silk fibroin/chitosan composite scaffold.

The combination of gelatin and sericin is capable of binding to cells, promoting cell proliferation in fibroblasts and keratinocytes. In vitro studies demonstrated that GS scaffolds had significantly increased cell proliferation when compared to non-scaffold models. Sericin has been found to enhance the migratory properties and adhesiveness of mammalian cells by activating the c-Jun pathway. It has also been suggested that it can increase cellular proliferation via a mechanism that has not yet been fully characterized, showing mitogenic properties. The GS scaffold promotes the clearance of cytotoxic reactive oxygen species, which can inhibit angiogenesis, and confers protection against oxidative stress to stem cells. Meanwhile, laminin, a component of the endothelial basal lamina, accelerates neovascularization in the wound bed, thereby enhancing wound healing and treatment efficacy. In the study conducted by Tyeb and colleagues, the authors demonstrated on a rat model that the MSC therapy group with GSL scaffold exhibited the most favorable outcomes in terms of wound healing, highlighting the potential of this approach as a therapeutic strategy for treating wounds in diabetic patients [92].

A silk fibroin/chitosan composite scaffold was used by Wu et al. to transfer adipose-derived stem/stromal cells (ADSC) to the wound site in diabetic rats. In this trial, there were a total of two control groups: one received only ASC grafts applied to the lesion, while the other received no grafts at all. Epidermal growth factor (EGF), tumor growth factor beta (TGF-B), and vascular endothelial growth factor (VEGF) were among the substances that the study group secreted in a higher amount than the control. The graft-only group’s VEGF expression was considerably higher. These results provide further evidence supporting the use of such scaffolds for the administration of ADSCs (Figure 5) [93].

There have also been several attempts to deliver MSC to the site of a wound, with the addition of pharmacological molecules as stimulating factors. One of the more promising studies was conducted by Seo et al., who superficially applied Exendin-4 (Ex-4), a glucagon-like peptide-1 receptor agonist, to the wound area after injecting MSCs. Ex-4 is known to have beneficial effects on diabetes. In vitro angiogenesis assays demonstrated that treatment with Ex-4 and ADSC-conditioned media (CM) improved migration, invasion, and proliferation of human endothelial cells. Combining a cell transplant with a topical application of a diabetes drug did not result in a decrease in blood glucose. Nevertheless, concomitant administration of Exendin-4 (Ex-4) and ADSC-conditioned media (CM) resulted in a significantly superior therapeutic effect compared to the individual treatments alone. Proliferation assays revealed a marked increase in proliferation following treatment with Ex-4, ADSC-CM, or the combination of both. The investigation demonstrated an increase in vascular endothelial growth factor (VEGF) expression in the wound area following treatment with Exendin-4 (Ex-4), ADSCs, or their combination. Nonetheless, co-administration of Ex-4 and ADSCs did not elicit a further enhancement in VEGF expression [94].

The systemic administration of MSC remains poorly understood, with limited clinical trials exploring this route for treating diabetic foot ulcers. Unlike in systemic diseases like, for example, scleroderma, where MSCs are widely distributed and have been shown to halt internal fibrosis [95,96].

One such study, conducted by Yan et al., sought to directly compare intravenous (IV) administration with topical administration of MSCs in a rat model. Their findings shed light on the comparative efficacy and safety of these two modalities. In the research, wounds treated with bone marrow-derived MSC showcased faster healing rates in both the topical and intravenous (IV) administration groups compared to the control group throughout the study period (*p* < 0.05). Notably, the IV group exhibited a significantly superior healing rate compared to the topical group at days 3 (*p* < 0.01) and 10 (*p* < 0.05). Additionally, histological examination revealed complete epithelialization in both the topical and IV groups by day 14, indicating the effectiveness of both administration methods in facilitating wound closure. These studies also have detected systemically administered cells in various organs such as the lungs, pancreas, liver, kidneys, and the wound site. Moreover, the study unveiled supplementary advantages linked with systemic transplantation of BM-MSCs, such as the amelioration of hyperglycemia. These findings highlight the potential systemic effects of MSC therapy beyond wound healing, further emphasizing the importance of considering both local and systemic implications when choosing the administration route [97].

## 8. Summary

The utilization of MSC represents a propitious therapeutic avenue for wound healing. Nevertheless, further investigation is warranted to refine the technique of administration and to discern the optimal milieu for cellular delivery. Although the outcome of current research is encouraging, they are not suitable enough to determine whether this treatment modality is effective. Recent clinical trials investigating the use of MSCs in chronic wounds, such as DFU, have explored both autologous and allogeneic MSCs. Autologous MSCs, derived from the patient’s own tissue, have demonstrated promising results in accelerating wound healing and reducing ulcer size. Studies, such as Tanios et al., reported significantly faster healing rates with autologous adipose-derived MSCs compared to standard care [98]. Meanwhile, trials with allogeneic MSCs, where cells are donated from another individual, have shown similar positive outcomes. For example, Uzun et al. reported the safe and effective use of allogeneic MSCs, with no significant immune reactions observed [99]. Both types of MSC therapies, autologous and allogeneic, exhibited enhanced healing capabilities in chronic wounds, though more research is needed to address long-term safety and efficacy. Therefore, future research ought to concentrate on improving MSC delivery to the wound site and investigating the underlying mechanisms by which we can maximize MSC’s regenerative potential. It is believed that via continued research and development efforts, MSC-based therapies may be more widely adopted in the clinical setting, improving the clinical outcomes of patients with chronic wounds.

## Figures and Tables

**Figure 1 ijms-25-12171-f001:**
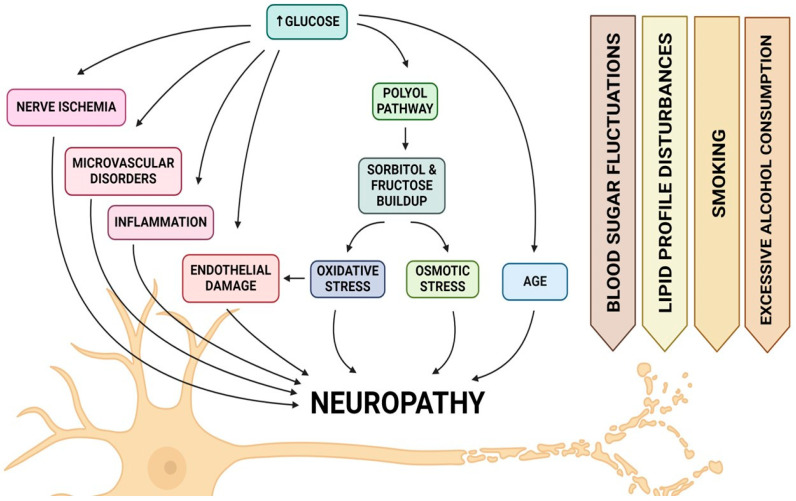
Ilustration of pathophysiological factors that contribute to the development of neuropathy in diabetic foot syndrome.

**Figure 2 ijms-25-12171-f002:**
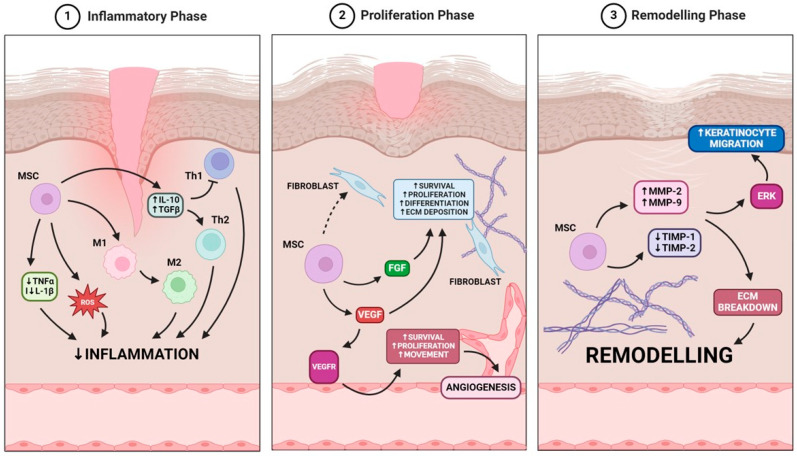
MSCs role in wound healing is based on the modulation of three key phases: the inflammatory phase, where MSCs reduce inflammation via cytokine regulation and macrophage polarization; the proliferation phase, where MSCs enhance fibroblast activity and angiogenesis through VEGF and FGF signaling; and the remodeling phase, where MSCs regulate extracellular matrix breakdown and keratinocyte migration to promote tissue regeneration.

**Figure 3 ijms-25-12171-f003:**
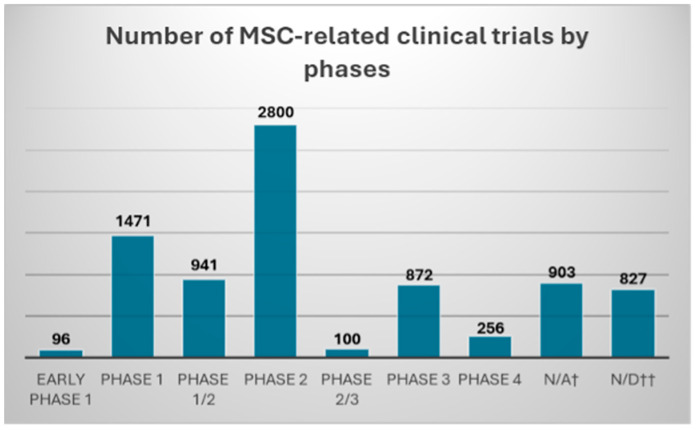
Number of MSC-related clinical trials found in clinicaltrials.gov database with “cell therapy” keyword, and “completed” filter. Clinical trials were divided according to the declared trial phase into: early phase 1; phase 1/2; phase 2; phase 2/3; phase 3; and phase 4. †—“N/A” trial phase information was deemed “not applicable” in clinicaltrials.gov; ††—“N/D” no data were provided regarding the trial phase in clinicaltrials.gov database.

**Figure 4 ijms-25-12171-f004:**
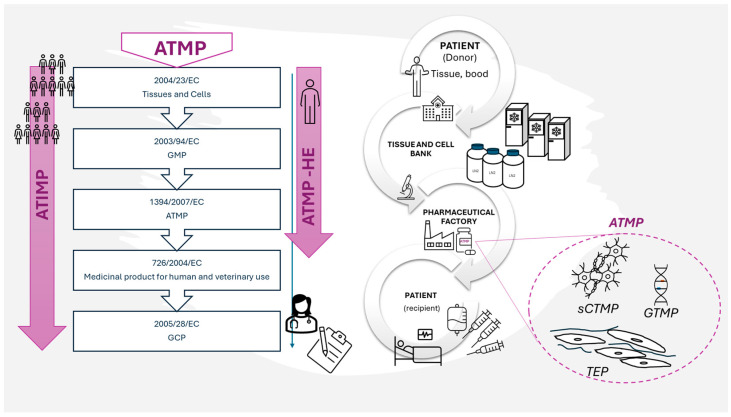
An exemplary route of biological material for ATMP in connection with legal regulations. Only the most fundamental regulations are indicated.

**Figure 5 ijms-25-12171-f005:**
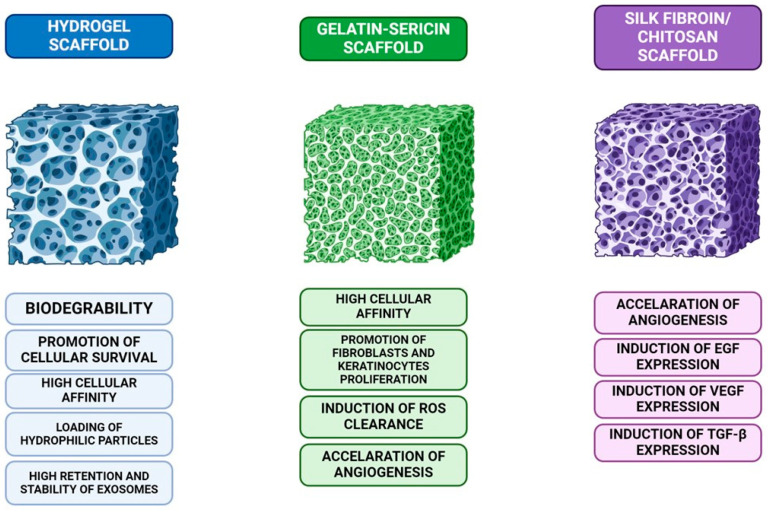
A comparison of three different scaffolds used in tissue engineering: hydrogel scaffolds, known for their biodegradability, cellular survival promotion, and exosome stability; gelatin-sericin scaffolds, which offer high cellular affinity, induce reactive oxygen species (ROS) clearance, and enhance fibroblast and keratinocyte proliferation; and silk fibroin/chitosan scaffolds, which accelerate angiogenesis and stimulate the expression of growth factors like EGF, VEGF, and TGF-β, crucial for tissue regeneration.

**Table 1 ijms-25-12171-t001:** Completed clinical trials listed in the clinicaltrials.gov database, which uses cells in the therapy of diabetic foot syndrome.

NCT Number	MSC Type	Phases	Completion Date
NCT01065337	BMSC	PHASE2	1 February 2009
NCT02619877	ADSC	PHASE2	1 October 2016
NCT02070835	other	N/A †	1 December 2022
NCT03183804	ADSC	N/D ††	24 October 2018
NCT06373809	PDSC	EARLY_PHASE1	31 March 2024
NCT02092870	ADSC	PHASE2	9 September 2019
NCT02799121	other	PHASE4	29 March 2019
NCT03881254	other	N/A †	28 July 2021
NCT03276312	ADSC/Adipose tissue	N/A †	1 March 2018
NCT00536744	other	PHASE3	1 September 2010
NCT04255004	other	N/A †	1 December 2019
NCT04633642	other	N/A †	31 December 2019
NCT03183726	ADSC	N/D ††	31 July 2017
NCT03865394	ADSC	PHASE1|PHASE2	30 September 2021
NCT01596920	other	PHASE4	1 March 2014
NCT00872326	BMSC	PHASE1|PHASE2	1 May 2009
NCT00987363	BMSC	PHASE1|PHASE2	1 March 2013
NCT02224742	other	PHASE4	1 May 2018
NCT02329366	other	N/D ††	1 December 2020
NCT03547635	other	N/A †	26 February 2019
NCT03636867	other	N/D ††	31 August 2020
NCT03267784	other MSC	PHASE1|PHASE2	29 June 2020
NCT01232673	BMSC	PHASE2	1 December 2010
NCT01113307	other	N/D ††	1 September 2011

†—trial phase information was deemed “not applicable” in clinicaltrials.gov. ††—no data were provided regarding the trial phase in clinicaltrials.gov database.

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
