# Peer review of "The Potential of Mesenchymal Stem/Stromal Cells in Diabetic Wounds and Future Directions for Research and Therapy—Is It Time for Use in Everyday Practice?"

_ijms, 2024, doi:10.3390/ijms252212171_

Round 1
Reviewer 1 Report
Comments and Suggestions for Authors
The authors present a review on mesenchymal stem cells in for clinical use in chronic wound therapy.
An introduction is followed by a lengthy description of chronic wound (healing) and its pathophysiology.
This is followed by the role of MSC in wound healing, a paragraph on legal regulations in Europe, allogenic products' advantages and challenges, practical issues related to ATMP release and ways of administration of MSC.
The title of the manuscript “The potential of mesenchymal stem/stromal cells in diabetic wounds and future directions for research and therapy - is it time for use in everyday practice? is somewhat misleading”.
While the authors describe in the third paragraph “ MSC role in wound healing phases” some effects of MSC it is unclear to the reader whether these findings have been made in wound healing studies (either human or animal) in vivo or whether these findings result from in vitro studies showing some potential effects . Every information in this paragraph has to been screen and edited for in vivo versus in vitro.
Furthermore a paragraph on the definition of MSC and its specific expression patterns should be included. A remark on the difference on pure MSC versus MSC with stromal fraction should be included. This is of great importance for the reader to correctly understand clinical studies on MSC and fat derived cell concentrates.
If the authors want to keep the part of the title “is it time for use in everyday practice” they need to include a paragraph on RCT with MCS in chronic wounds. This should include autologous MSC studies as well as allogeneic MSC studies.
Line 408 Are there any in vitro studies on MSC viability after thawing . Please search the literature.
Line 436 Pleases clarify if the dosage mentioned is for chronic wounds as local therapy ? And if so for which size of wounds??
As the legal bodies were cautious about the cancerogenic potential of MSC, please discuss this aspect in your manuscript.
In “7. The ways of administration of MSC to intended locations” information on the type of studies are missing. Most of the cited studies are animal studies. Please include this information for every statement.
Author Response
Thank you for your valuable feedback.
Comment 1: While the authors describe in the third paragraph “ MSC role in wound healing phases” some effects of MSC it is unclear to the reader whether these findings have been made in wound healing studies (either human or animal) in vivo or whether these findings result from in vitro studies showing some potential effects . Every information in this paragraph has to been screen and edited for in vivo versus in vitro.
Response 1: We have carefully evaluated the third paragraph discussing the role of MSCs in wound healing phases, as you suggested. Each piece of information has been thoroughly screened and edited to clarify whether the findings are from in vivo (human or animal wound healing studies) or in vitro studies demonstrating potential effects. We have explicitly indicated which studies are in vivo and which are in vitro throughout the paragraph. The corresponding changes have been made to the text to improve clarity for the reader.
Comment 2: Furthermore a paragraph on the definition of MSC and its specific expression patterns should be included. A remark on the difference on pure MSC versus MSC with stromal fraction should be included. This is of great importance for the reader to correctly understand clinical studies on MSC and fat derived cell concentrates.
Response 2:
A paragraph “MSCs are multipotent cells capable of differentiating into adipocytes, osteoblasts, and chondrocytes. They are identified by positive expression of CD73, CD90, and CD105, and the absence of hematopoietic markers like CD34 and CD45. MSCs can be derived from various tissues, including bone marrow and adipose tissue, and exhibit heterogeneity based on their origin​. In clinical contexts, pure MSCs refer to isolated, well-characterized cell populations that meet specific phenotypic and functional criteria. In contrast, MSCs in the stromal vascular fraction (SVF) are part of a mixed cell population, containing other cell types like endothelial and immune cells. This distinction is critical for interpreting clinical outcomes, as pure MSCs offer targeted effects, whereas SVF-derived therapies involve multiple cell interactions, potentially complicating the understanding of therapeutic efficacy​.” was added in lines 47-57.
Comment 3: If the authors want to keep the part of the title “is it time for use in everyday practice” they need to include a paragraph on RCT with MCS in chronic wounds. This should include autologous MSC studies as well as allogeneic MSC studies.
Response 3: „Recent clinical trials investigating the use of MSCs in chronic wounds, such as DFU, have explored both autologous and allogeneic MSCs. Autologous MSCs, derived from the patient’s own tissue, have demonstrated promising results in accelerating wound healing and reducing ulcer size. Studies, such as Tanios et al., reported significantly faster healing rates with autologous adipose-derived MSCs compared to standard care. Meanwhile, trials with allogeneic MSCs, where cells are donated from another individual, have shown similar positive outcomes. For example, Uzun et al. reported safe and effective use of allogeneic MSCs, with no significant immune reactions observed. Both types of MSC therapies, autologous and allogeneic, exhibit enhanced healing capabilities in chronic wounds, though more research is needed to address long-term safety and efficacy​.” was added in lines 628-638.
Comment 4: Line 408 Are there any in vitro studies on MSC viability after thawing . Please search the literature.
Response 4: We have searched the literature and "François et al. showed that the percentage of viability MSC collected from continuous culture was about 90%, while their freshly thawed counterparts dropped to about 40%. They also found a deterioration in the immunosuppressive properties of the tested cells (76). Some authors even suggest that the poorer properties of the cells after thawing may be the reason for the unsatisfactory results of clinical trials with their participation (77-79). Therefore, it is recommended to consider this important aspect already during the planning of the manufacturing process, creation of the new product specification, and later during the testing of cell parameters at the stage of application of the finished product. This will allow us to avoid the high costs of clinical trials due to suboptimal use of the potential of MSC cells during manufacturing and the risk of interference resulting from the impact of the thawing process itself." was added in lines 448-458.
Comment 5: Line 436 Pleases clarify if the dosage mentioned is for chronic wounds as local therapy ? And if so for which size of wounds??
Response 5: "(...) applied both systemically or locally per injection or directly on wounds" was added in line 483.
Comment 6: As the legal bodies were cautious about the cancerogenic potential of MSC, please discuss this aspect in your manuscript.
Response 6: "One of the concerns expressed by regulatory bodies is the supposed cancerogenic potential of MSC. Therefore, EMA guidelines were issued in 2024 regarding the requirements for preclinical ATIMP studies involving extensive tumorigenic and immunogenic safety assessment (EMA/CAT/123753/2024). However, MSCs show almost negligible immunogenicity and by some are even called immune-evasive (84, 85). Moreover, unmodified MSCs, that are often used in cell therapies, which underwent only minimal in vitro manipulations such as isolation and expansion retain their genetic stability. The reported rare tumorigenic events were not caused by MSC but by the ATIMP contamination with tumor cells (86). Therefore, legal bodies have put especially high pressure for researchers to ascertain the identity of cells used in ATIMP as the origin and purity of starting cellular material is one of the key factors in the safety of the ATIMP." was added in lines 485-495.
Comment 7: In “7. The ways of administration of MSC to intended locations” information on the type of studies are missing. Most of the cited studies are animal studies. Please include this information for every statement.
Response 7: The relevant mentions regarding the type of studies have been added in paragraph “The ways of administration of MSC to intended locations.” We have specified which studies are based on animal models for each statement to ensure clarity of the presented information.
Reviewer 2 Report
Comments and Suggestions for Authors
The manuscript by Sieńko et al. addresses the current issue of modulating the healing of chronic (diabetic) wounds using MSCs. This topic is certainly timely and interesting for readers. I would recommend the article for acceptance after "major revision."
The main issues with the manuscript are summarized in the following points:
- A large percentage of the citations are review articles, with only a small proportion of primary sources. This proportion is better in the sections of the text directly addressing the use of MSCs, which is commendable. Nevertheless, I would suggest revising the references in other parts of the text as well and aiming for primary sources where possible.
- The text on lines 49-53 lacks citations, specifically regarding the claim about clinical studies.
- I would recommend shortening/revising Chapter 2 (sections 2.1-2.3), where the information is relatively repetitive and quite general. The text could be significantly condensed, and references could be used more effectively.
- The healing process is traditionally divided into four phases: besides hemostasis, there are the inflammatory, proliferative (granulation), and remodeling phases, which are reflected in chapters 3.1-3.3. However, Chapter 3.4, "Angiogenesis," appears unusual because angiogenesis is not a phase of healing in itself but a process primarily occurring during the proliferative phase. I would either include it within this phase or separate it in some other way in the text to avoid the impression that it is the fourth phase of healing after remodeling.
- What is the specific feature of the only registered MSC-based product mentioned, Alofisel (line 296)? Could additional information be provided that might be interesting to readers?
- Lines 298-300 – what specific MSC-related clinical trials were found, and what phase are they in? Could a summary table with this information be added?
- Regarding Figure 4 – why were these specific scaffolds selected? The literature contains many other materials tested in connection with MSC therapy. Could scaffolds used in clinical trials be selected, for example? The entire Chapter 7 feels inconsistent; the individual paragraphs do not flow well, and the selection of referenced papers seems completely random.
- It might be an issue with the automatically generated PDF file, but the images are of poor quality and unreadable.
Author Response
Comments 1: A large percentage of the citations are review articles, with only a small proportion of primary sources. This proportion is better in the sections of the text directly addressing the use of MSCs, which is commendable. Nevertheless, I would suggest revising the references in other parts of the text as well and aiming for primary sources where possible.
Response 1: Thank you for your constructive feedback. In response, we have incorporated additional original research articles across many citations to strengthen the proportion of primary sources. However, we have retained some review articles as accompanying references due to the valuable context and significant information they provide.
Comments 2: The text on lines 49-53 lacks citations, specifically regarding the claim about clinical studies.
Response 2: Six new citations (9-15) were added to confirm sentences in mentioned fragment of text.
Comments 3: I would recommend shortening/revising Chapter 2 (sections 2.1-2.3), where the information is relatively repetitive and quite general. The text could be significantly condensed, and references could be used more effectively.
Response 3: Thank you for your helpful suggestion. We have revised and condensed Chapter 2, specifically sections 2.1–2.3, by removing redundant data and citations that contributed to repetitive content. This has streamlined the section and improved the focus and effectiveness of the references used.
Comments 4: The healing process is traditionally divided into four phases: besides hemostasis, there are the inflammatory, proliferative (granulation), and remodeling phases, which are reflected in chapters 3.1-3.3. However, Chapter 3.4, "Angiogenesis," appears unusual because angiogenesis is not a phase of healing in itself but a process primarily occurring during the proliferative phase. I would either include it within this phase or separate it in some other way in the text to avoid the impression that it is the fourth phase of healing after remodeling.
Response 4: Thank you for your insightful comment. We have addressed this by combining the "Angiogenesis" paragraph with the "Proliferative Phase" to avoid any misinterpretation that angiogenesis represents a distinct, fourth phase of healing. This integration should clarify its role as a process within the proliferative phase.
Comments 5: What is the specific feature of the only registered MSC-based product mentioned, Alofisel (line 296)? Could additional information be provided that might be interesting to readers?
Response 5: Thank you for your comment. "It illustrates the wider problem concerning the ATMPs’ development. First of all, the requirements of the regulatory bodies are identical, i.e. equally demanding for different stages of the product development. Therefore, validation of each production stage must be carried out in accordance with many strict regulations at a very early stage of ATMP development. In the case of the standard medicinal, there are less regulations to be watched and the general methodology is in principles the same – typical for chemical substances. In the case of advanced therapy medicinal products (ATMPs), the validation plan is in most cases individual, which is associated with a significantly greater investment of time and costs, and is always associated with a longer path to regulatory approval and is burdened with the risk of additional requirements. What's more, the EMA recommendations dedicated to ATMPs change dynamically. Just to mention, the “Guideline on quality, non-clinical and clinical requirements for investigational advanced therapy medicinal products in clinical trials” (EMA/CAT/123573/2024), currently required by the Clinical Trials Registration Offices, was announced on March 2024 as a draft with no date for coming into effect, while the previous version, of the January 2019 (EMA/CAT/852602/2018), remained at the draft stage with no date for coming into effect as well. Those guidelines are accompanied by many other obligatory recommendations - frequently changing and not always entirely consistent with each other. Taking into account, that because of the species-determined active substance, the preclinical proof of concept based on animal observations is far from the real situation, pharmaceutical companies are not interested in investing in ATIMP development at early stages of clinical trials. Therefore, in this phase, the financial and organizational burden falls on the shoulders of the research centers, which is a serious obstacle to ATIMP being introduced to the market." was added in lines 294-317.
Comments 6: Lines 298-300 – what specific MSC-related clinical trials were found, and what phase are they in? Could a summary table with this information be added?
Response 6:Thank you for your suggestion. We have added Figure 3, which summarizes the phases of the MSC-related clinical trials identified, as well as Table 1, which details clinical trials specifically using MSCs in the treatment of diabetic foot ulcers. These additions should provide a clearer overview of the current status and focus of MSC clinical applications.
Comments 7: Regarding Figure 4 – why were these specific scaffolds selected? The literature contains many other materials tested in connection with MSC therapy. Could scaffolds used in clinical trials be selected, for example? The entire Chapter 7 feels inconsistent; the individual paragraphs do not flow well, and the selection of referenced papers seems completely random.
Response 7: Thank you for your insightful feedback on my manuscript, particularly regarding the section on scaffold selection for mesenchymal stem cell administration. We appreciate your concern about the selection process appearing somewhat random, and we apologize if our choice of studies and scaffolds seemed unclear or lacking in systematic criteria. While our selection of presented scaffolds was indeed subjective, we aimed to include the most relevant and well-documented scaffolds, specifically focusing on those that have demonstrated a significant impact on the regenerative potential of mesenchymal stem cells in wound healing applications. The selected studies reflect current advancements in scaffold types with detailed characterizations, ensuring they are not only viable for cell adherence and viability but also that they effectively enhance regenerative outcomes. Our intent was to emphasize those that show promise in enhancing therapeutic efficacy. We clarified these criteria in the text to provide a clearer rationale for each choice by adding additional citations. I hope this will make the section more transparent and aligned with the manuscript’s purpose.
Comments 8: It might be an issue with the automatically generated PDF file, but the images are of poor quality and unreadable.
Response 8: Thank you for bringing this to our attention. The graphics were created using BioRender, and we will be providing the original, high-resolution images directly to the journal to ensure clarity and readability in the final publication.
Round 2
Reviewer 1 Report
Comments and Suggestions for Authors
The manuscript has significantly been improved. All queries have been sufficiently been answered. I have no further queries.
Reviewer 2 Report
Comments and Suggestions for Authors
I appreciate the changes made based on my previous comments. In my opinion, the manuscript is now acceptable for publication.